# The Role of Sustainability Reporting in Reducing Information Asymmetry: The Case of Family- and Non-Family-Controlled Firms

Abdul Rahman Al Natour [1], Rasmi Meqbel [2,*], Salah Kayed [2] and Hala Zaidan [3]

1 Department of Accounting, Faculty of Administrative & Financial Sciences, University of Petra, Amman 11196, Jordan; abdulrahman.alnatour@uop.edu.jo

2 Department of Accounting, Faculty of Economic and Administrative Sciences, The Hashemite University, Zarqa 13133, Jordan; salahk@hu.edu.jo

3 Department of Accounting, School of Business, The University of Jordan, Amman 11942, Jordan; h.zaidan@ju.edu.jo

* Correspondence: rasmi.meqbel@hu.edu.jo

**Abstract:** This study aims to examine the link between sustainability reporting and information asymmetry in family- and non-family-controlled firms for a sample of 641 UK firms listed in the FTSE all-share index during the period 2010–2017. The findings show a negative and significant relationship between sustainability reporting and IA. The results also show that the sustainability reporting–information asymmetry nexus is weaker in family-controlled firms. The findings of this study should improve our understanding of sustainability reporting motivations, particularly in companies that are controlled by families. Moreover, an explanation of the role of family-controlled firms in mitigating or exacerbating this relationship will surely help the British regulators improve corporate governance rules related to various ownership structures. For policy makers, it is important to confirm that sustainability reporting is representative of actual corporate activities and is not only used to mislead stakeholders.

**Keywords:** corporate social responsibility disclosures; sustainability reporting; information asymmetry; bid–ask spread; family-controlled firms

## 1. Introduction

In recent years, sustainability reporting has become an area for market participants [1]). Various stakeholders rely on sustainability reporting and CSR disclosure to inform their decision-making processes and forecasting [2,3]. A recent international survey conducted by KPMG showed that 78 percent of world's 250 largest companies (based on the Fortune 500 ranking for the year 2016) include CSR information in their annual reports, believing that sustainability reporting is relevant for their existing and potential investors (KPMG, United Nations Environment Programme, Global Reporting Initiative, University of Stellenbosch (2010): *Carrots and Sticks—Promoting transparency and sustainability: An Update on Trends in Voluntary and Mandatory Approaches to Sustainability Reporting*). Studies that have examined the benefits of CSR engagement by linking it with direct measures of corporate financial performance find that its application can increase the attention paid to it by analysts [4], resulting in more accurate forecasting [3] and, consequently, favourable recommendations [5].

It can be argued that a company's information is one of the most important determinants of equity risk, company performance, and cost of capital [1,6,7]. According to [1,8], a company's disclosures can help reduce information access differences between managers and shareholders, thus increasing market liquidity, reducing the volatility of stock prices, and decreasing the company's equity capital costs e.g., [9–12]. In contrast, an unpredictable environment causes business friction by introducing adverse selection, which leads to

lower liquidity and a higher cost of company capital [13]. Despite the importance of information asymmetry (IA) in capital markets and corporate decision making [14], studies of the association between firms' disclosures and IA generally focus on the effect of financial disclosures on IA. However, little is known about whether, and in what way, sustainability reporting (a type of non-financial disclosure) influences IA [3,6,15,16], an area of business activity that is becoming increasingly attractive for market participants [15].

Cormier and Ledoux (9) and Dhaliwal and Radhakrishnan (2) were the first to attempt to examine the influence of CSR disclosures on IA. The authors of [17] argue that environmental and social disclosures substitute for each other in mitigating stock market asymmetry, while the authors of [3] found that CSR reports were linked to lower forecast errors by analysts, and that this association was moderated by stakeholder orientation and financial transparency. Other studies have investigated these concerns in a much broader way, by considering liquidity [18] or the cost of capital [19] as proxies for IA. The majority of research, however, focuses on CSR performance rather than the level of sustainability reporting [15,16,20]. Accordingly, this study aims to investigate the effects of sustainability reporting on market participants by examining the link between CSR disclosures and IA, taking into account the influence of each component of sustainability reporting (such as environmental, social, and governance—hereafter ESG). It also addresses how family-controlled firms, as an example of informed investors, could moderate the sustainability reporting–IA nexus.

Overall, this study contributes to the existing literature on sustainability reporting, IA, and family-controlled firms in several ways. First, unlike most previous studies, which use a single indicator of CSR (performance and/or disclosure), the current study tests the relationship between CSR and IA using both as single CSR indicator that combines three elements (environmental, social, and governance (ESG) score); it then tests the effect of each element separately. The reason for using separate CSR indicators is that sustainability reporting is a multidimensional concept that represents the relationship between business and society [21,22]. Despite its complexity, several studies have dealt with it as a homogeneous activity and regressed it as a single indicator without considering its individual dimensions e.g., [6,15,16,23]. Rationally, considering sustainability reporting dimensions separately (e.g., environmental, social, and governance scores) could result in a better understanding of management strategies and their mentality towards CSR activities.

Second, the moderating role of the relationship between sustainability reporting and IA has been previously studied by considering firm characteristics [6]; equity risk [16]; and institutional ownership [15]. This study builds on the previous literature by examining the moderating role of family-controlled firms in the CSR–IA nexus. The reason for choosing such firms is that they are argued to have unique characteristics that distinguish them from others [24,25]. Moreover, family ownership is considered to be the most prevalent type of ownership around the world [26–28]. Third, this study extends the methodological approaches of previous studies by using the generalised method of moments (GMM) model. The causal association between sustainability reporting and IA could be endogenous as a result of managerial policies and other factors that result in simultaneity and reversed causality. Therefore, if sustainability reporting and IA are simultaneously determined, the ordinary least squares (OLS) method will not be accurate. Consequently, based on the work of Arellano and Bond [29], the GMM model is considered useful in addressing these issues and controlling for heterogeneity.

Based on a sample of UK firms listed on the FTSE All-Share Index during the period 2010–2017, our findings show that the relationship between sustainability reporting and IA is negative and significant, suggesting that sustainability reporting can play a complementary role in reducing the information gap that exists between firms and their stakeholders. The results also show that the relationship between sustainability reporting and IA is weaker in family-controlled firms, as it tends to be positive. This means that family-controlled firms, with their information advantage, may disguise their trading through small transactions in order to maximise their profit by buying at lower asking

prices and selling at higher prices. Moreover, the authors of [30] suggest that informed investors can adjust their portfolios due to the private information they possess, whereas less-informed investors are not able to adjust their portfolios effectively due to their lack of private information, which exacerbates IA by increasing the risks faced by less-informed investors and consequently widens the bid–ask spread. Finally, by independently examining the influence of the environmental, social, and governance elements on IA, we found that the negative relationship remained similar to our previous findings. However, an investigation of the interaction between family-controlled firms and each of these pillars (environmental, social, and governance) revealed different tendencies. For example, we found that family-controlled firms weaken the negative influence of environmental and governance disclosure scores on IA but strengthen the negative influence of social disclosures on IA. Generally, this outcome indicates that family-controlled firms tend to be selective in sustainability reporting, as there are certain governance and environmental issues that they tend to hide, which creates an adverse selection problem.

The rest of the study is structured as follows. Section 2 presents the literature review and hypothesis development with regard to the sustainability reporting–IA relationship, before moving on to further discuss the influence of family-controlled firms on the above relationship. We then discuss the sample and measurement of the main study variables, together with our research design. The final two sections present the empirical outcomes and discuss the findings.

## 2. Literature Review and Hypothesis Development

### 2.1. Sustainability Reporting and Information Asymmetry

Information asymmetry is a condition in which one party in a relationship has better access to information than the other party [31]. Accounting and finance studies have mainly discussed two types of IA. The first occurs as a result of the separation between managers (the "agents") and investors (the "principals") [8,9,32], whereas the second type arises between the investors themselves (i.e., informed and less-informed investors, who are a group of majority and minority shareholders) [33]. Although the latter type has attracted less attention in the literature, the following sections deal with both types of IA interchangeably, arguing that corporate disclosures, namely sustainability reporting, influence the information asymmetry problem. The second part of the discussion is directed towards the role of family-controlled firms, which are categorised as informed investors, in moderating the sustainability reporting–IA nexus.

The opportunities and challenges of IA are fundamentally related to several theories, including agency theory [32]. This theory, in its classical form, is based on the principal–agent relationship, in which managers ("agents") are appointed by a firm to act on behalf of shareholders ("principals") [32]. This kind of separation gives managers the privilege of enjoying better access to information about the firm's prospects, which they exploit in projects that serve their interests [32]; this constitutes a moral hazard problem. Such conflicts may lead to a collapse in the performance of the capital market, meaning that if shareholders cannot distinguish between "good" and "bad" business actions, managers committing "bad" actions will try to claim that they are "good" actions, and shareholders will value both "bad" and "good" actions at the same level. Therefore, the capital market will undervalue some "good" actions and overvalue others that are "bad" based on the information available to the managers [14], constituting an adverse selection problem [9,34]. Diamond and Verrecchia [8] and Leuz and Verrecchia [34] argue that IA can create such costs as a consequence of adverse selection, because information gathering by investors takes time and can therefore be expensive, raising the opportunity costs.

One of the key measures to narrow the information gap that might exist as a result of the separation between ownership and control is to keep investors informed by revealing information to the public in the form of corporate disclosures [8,35]. Logically, when investors have more information about a company's activities, they will be able to value the available alternatives and make accurate decisions [36]. In this vein, Diamond and

Verrecchia [8] argue that company disclosures can help reduce information differences between managers and shareholders, thus increasing liquidity in the market, reducing the volatility of stock prices, and decreasing companies' equity capital costs e.g., [9–12].

Information disclosures by firms can be issued using a set of communication reports, which may be mandatory, in the form of regulated reports and other periodic regulatory filings, or voluntary, meaning that they are not required by law or other regulatory bodies [14]. Another essential distinction which may be involved within mandatory and voluntary disclosures is the distinction between financial and non-financial disclosures, with the latter referring to social and environmental disclosures. Generally, financial disclosures are more likely to be mandatory, whereas non-financial disclosures tend to be less disciplined. In this vein, the theoretical literature shows that both voluntary and mandatory disclosures reduce information asymmetry [37]. However, there is little empirical evidence that proves whether, and in what way, sustainability reporting (an example of a non-financial disclosure) influences IA [3,6,15,16]. While the theoretical literature shows that both types of disclosure could help in reducing IA, this study emphasises information related to sustainability reporting, an area of business activity that is becoming increasingly attractive for market participants [15,38,39].

Theoretically, the relationship between sustainability reporting and information asymmetry can be explained from the perspective of stakeholder theory, according to which managers have a fiduciary duty towards all stakeholders instead of maintaining exclusive relationships with them [40] (stakeholders are any identifiable individual or group who can affect or be affected by the achievements of a firm's objectives (Freeman and Reed, 1983)). Meeting the expectations of different stakeholder groups by actively committing to CSR can help to improve a company's reputation [41]. Therefore, it has been argued that reputation building is linked with higher-quality earnings reporting [42], which ultimately reduces IA [43]. Previous studies argue that CSR is positively related to earnings reporting quality, suggesting that it creates an atmosphere that inspires managers to adopt a public-responsibility-oriented mentality, which subsequently encourages the issuance of more transparent financial reporting and meets stakeholder expectations [44–46]. Clarkson, Li [47] found that socially responsible firms tend to disclose more information to the public in order to build their reputation and inform stakeholders about their social responsibility [3,48]. In the same vein, when a company has built up a certain reputation (via sustainability reporting), it can improve its financial performance by attracting more qualified employees, boosting customer loyalty, and gaining considerable attention from analysts [49]—the so-called business case for sustainability [50].

Studies that have empirically examined the relationship between sustainability reporting and IA have generally found evidence of a negative relationship. For example, Cho, Lee [15] investigated the link between CSR performance and information asymmetry, relying on a bid–ask spread (the amount by which the ask price exceeds the bid price for an asset in the market) as a proxy for IA and considering a sample of the US stock market over the period 2003–2009. Their main finding was that both negative and positive CSR performance are negatively related to the bid–ask spread. More specifically, a negative CSR performance tends to be more effective than a positive performance in mitigating IA. This negative relationship was also observed by the authors of [6], who tested whether sustainability reporting reduced the bid–ask spread in a sample of 391 Australian non-financial companies during the period 2004–2014. This negative association was reported to be more prominent in larger companies and those that possessed stronger market power. Cui, Jo [16] recently provided evidence for the relationship between sustainability reporting and IA using a sample of US non-financial companies during the period 1991–2010. IA was measured using three different proxies: the dispersion of analysts' forecasts, the price impact measure, and the bid–ask spread. After employing two-stage least squares (2SLS) and generalised method of moments (GMM) models, they found that CSR was negatively related to IA. Dhaliwal, Radhakrishnan [3] further examined the impact of CSR on analysts' forecast accuracy. They found that issuing stand-alone CSR reports was positively related

to analysts' forecast accuracy, implying that sustainability reporting reduces IA (a negative relationship). In this respect, we propose the following hypothesis:

**Hypothesis 1 (H1).** *Sustainability reporting is negatively related to information asymmetry.*

*2.2. The Influence of Firm Ownership on Information Asymmetry*

While the separation between agents and principals is argued to be one of the key factors that lead to IA [14], investors are heterogeneous with regard to the level of information they possess or have access to. For instance, those who hold the majority of shares could gain better access to information than minority shareholders [51]. In such a case, there exist informed and less-informed investors. Family-controlled firms, which are characterised by majority ownership, are seen to be a common case of majority and minority shareholders. Research on corporate ownership e.g., [26,28,52] shows that family ownership is the most prevalent ownership type around the world. Several studies have documented the fact that family-owned companies represent over one third of large US listed companies [51,53] and more than 55 percent of smaller companies. The percentage is also high in Europe, at around 44.29 percent in 13 Western European countries [28,53].

This concentration provides family owners with the privilege of holding higher managerial positions and allows them to engage in day-to-day activities, offering them better access to information. Accordingly, the gatekeeper role that family-controlled firms play over management behaviour could result in better monitoring, thus mitigating the classical form of the agency problem (the "principal-agent problem") and diminishing the IA that stems from the separation between management and ownership [54–56]. Previous studies have addressed agency problems from different perspectives. A conflict of interests can arise between: shareholders (principals) and managers (agents)—type-I agency conflict (Jensen and Meckling, 1976); majority and minority shareholders—type-II agency conflict (Morck, Shleifer, and Vishny, 1989); and shareholders and stakeholders—type-III agency conflict. A commercial entity dominated by an ownership concentration corresponds to type-II agency conflict. One issue of concern is that the majority ownership uses its power and privileges in an opportunistic way to further its own interests at the expense of the interests of the minority. Since family-controlled firms possess more information than other investors and have the opportunity to be involved in management, a type-II agency problem between the family as the majority investor and the other minority investors may arise [57,58].

Based on the various impacts of informed investors mentioned above, it is possible that the influence of family-controlled firms on the relationship between sustainability reporting and IA could help to reduce overall IA, implying that the role of such firms is not only to monitor management behaviour, but also to take the initiative in being more active and involved in sustainability reporting activities. On the other hand, family-controlled firms could increase the information gap by taking advantage of the private information they have as major shareholders, meaning that the relationship between sustainability reporting and IA will be positive. Cho, Lee [15] describe the first theory as the information efficiency effect and the second as the adverse selection effect.

The information efficiency effect theory argues that the private information possessed by the majority of family-controlled firms allows them to actively participate in the market [15]. Therefore, their trading will help to disseminate more information to the market and encourage other "less-informed" investors to imitate their behaviour. Consequently, stock market liquidity will increase and the bid–ask spread will be reduced [59,60]. According to the information efficiency theory, family-controlled firms will improve market liquidity by conducting sustainability reporting, publishing their information in a timely and detailed manner. Based on this reasoning, it is argued that the higher the proportion of family stock ownership, the stronger the negative relationship between sustainability reporting and IA.

Additionally, several previous studies have argued that the adverse selection of informed investors could increase information differences between informed and less-informed investors and thus widen the bid–ask spread [15]. Informed investors, with their information advantage, may disguise their trading through small transactions in order to maximise their profit by buying at lower ask prices and selling at higher bid prices. This method of trading can be sustained until any private information is fully disclosed to the public, or as long as the profit from trading against less-informed investors is adequate to cover any cost of information acquisition [61]. Moreover, Easley and O'hara [30] suggest that informed investors can adjust their portfolios using the private information they possess, whereas less-informed investors cannot, due to their lack of private information; this will increase the IA by raising the risks faced by less-informed investors, and, consequently, the bid–ask spread will be widened. Accordingly, a higher proportion of family ownership is expected to attenuate any reduction in IA attributed to sustainability reporting.

**Hypothesis 2 (H2).** *Family-controlled firms moderate the relationship between sustainability reporting and information asymmetry.*

### 3. Research Design

*3.1. Sample Selection and Data Sources*

The study sample consisted of UK companies listed on the FTSE All-Share Index over the period 2010 to 2017. This is a capitalisation-weighted index, representing around 98% of the market capitalisation of listed shares in the UK, combining the FTSE Small Cap, FTSE 100, and FTSE 250 indices. We chose the FTSE All-Share Index to capture a larger number of family firms from different industries and at different levels. Financial institutions were excluded from the sample due to their different nature and associated regulations related to social and environmental disclosures [62–65].

Data on the IA index, sustainability reporting, and financial variables were mainly collected from the Bloomberg database, while ownership data for family firms were collected from the FAME database. We dealt with missing data, especially ownership data, by examining firms' annual reports and their websites.

*3.2. Variable Measurement*

3.2.1. Information Asymmetry

Predicting IA is a complex task, since it deals with an unobservable phenomenon. Therefore, previous studies have introduced several proxies to measure IA, including bid–ask spread, stock liquidity (trading volume), price volatility, market-to-book ratio, the accuracy of analyst forecasts, and price impact measures [34,66,67]. In this study, the bid–ask spread (SPREAD) was used to measure IA; specifically, the annual average percentage of the daily bid–ask spread to the closing price. The wider the spread, the higher the degree of IA [68].

3.2.2. Family Firms

To distinguish between family- and non-family-controlled firms, studies have relied on various measurements and thresholds. Some e.g., [53,69,70] recognise family-controlled firms as those of which the family owns more than 20 percent of the shares. On the other hand, a 10% cut-off point has been used by other studies to define such firms [26,28,71–75].

Using a definition by Anderson and Reeb [76], a firm can be considered a family-controlled entity when family ownership exceeds 5% and/or there are two or more board members from the family (such entities are recognised as family firms even if they are not owned by one family). This definition has been used by several studies e.g., [57,77–79]. Family relationships include father, mother, sisters, brothers, sons, daughters, spouses, in-laws, aunts, nieces, nephews, and cousins [77]. Similarly, Claessens, Djankov [27] and Peng and Jiang [80] define firms as being family-controlled if family groups dominate more than 5% of voting rights (a family group could be one family or groups of families). In

addition, Chrisman and Patel [81] define a family firm as one in which family ownership exceeds 5% of the capital, and in which at least one family member serves as a member of top management.

This study relies on the definition of Anderson and Reeb [76] and Martin, Campbell [79], who classify a firm as a family-controlled entity when family ownership exceeds 5% and/or there are two or more board members from the family (these entities are recognised as family firms even if they are not totally owned by one family). A dummy variable equal to 1 was applied to family-controlled firms, and 0 otherwise.

### 3.2.3. Sustainability Reporting Measurement

To measure sustainability reporting, we used ESG disclosure scores provided by Bloomberg as a proxy for environmental, social, and governance disclosure levels in each firm.

The Bloomberg ESG database is a comprehensive index related to the environmental, social, and governance (ESG) disclosure of about 11,500 companies in more than 83 countries. For each company, Bloomberg developed key performance indicators (KPIs) and ratios, thus contributing to better analysis and comparison of companies regarding ESG metrics. The database also takes into consideration the differences between industry-specific factors. Accordingly, each company receives different ESG factors and is evaluated according to the industry it belongs to. For instance, points are earned for phone/mobile recycling only by telecommunications systems.

ESG disclosures are scored based on 100 ESG points (60 environmental (E), 26 social (S), and 14 governance (G)), with a score of 1 being granted to companies that disclose a minimum number of points and a score of 100 to those that fully disclose all points. If any of the 100 points is not disclosed and/or the company is not covered by an ESG group, 'N/A' will be indicated. As mentioned, each data point is weighted according to its importance; for example, greenhouse emissions have a greater weight than other disclosures. In addition, each company is evaluated in comparison to other companies in the same industry. This means that Bloomberg analysts do not derive or derivate data; they are mainly collected from company filings (e.g., annual reports, CSR sustainability standalone reports, firms' websites, and ESG surveys prepared by Bloomberg that are attached to the companies on an annual basis).

### 3.2.4. Empirical Model

The main model to test H1 was largely derived from Cho, Lee [15] and Cui, Jo [16], as follows:

$$SPREAD = \beta_0 + \beta_1 ESG\_SCORE_{it} + \beta_2 SIZE_{it} + \beta_3 LEV_{it} + \beta_4 ROA_{it} + \beta_5 GROWTH_{it} + \beta_6 ANALYSTS_{it} + \varepsilon_{it} \quad (1)$$

where *SPREAD* is the annual average percentage of the daily bid–ask spread to the closing price; *ESG_SCORE* is the total *CSR* score of the three pillars (environment, social, and governance); *SIZE* is the natural logarithm of total assets; *LEV* is a leverage ratio measured as long-term debt scaled by total assets; *ROA* is the return on assets ratio, measured as income before extraordinary items and scaled by lagged total assets; *GROWTH* is the sales growth rate; and *ANALYSTS* is the number of analysts following the company.

To test H2, we altered our study model to include the moderating effect of family ownership on the association between *SPREAD* and *ESG_SCORE*, as follows:

$$SPREAD = \beta_0 + \beta_1 ESG\_SCORE_{it} + \beta_2 Family_{it} + \beta_3 ESG\_FAMILY_{it} + \beta_4 SIZE_{it} + \beta_5 LEV_{it} + \beta_6 ROA_{it} \\ + \beta_7 GROWTH_{it} + \beta_8 ANALYSTS_{it} + \varepsilon_{it} \quad (2)$$

where Family is a dummy variable equal to 1 for family-controlled firms and 0 otherwise (see family-firm criteria in the variable measurement section) and *ESG_Family* is the interaction between family-controlled firms and *ESG* score. The causal association between sustainability reporting and IA could be endogenous as a result of managerial policies and

other factors that result in simultaneity and reversed causality. Therefore, if sustainability reporting and IA are simultaneously determined, the ordinary least squares (OLS) method will not be accurate. Consequently, based on Arellano and Bond [29], the GMM model was considered useful in addressing these issues and controlling for heterogeneity.

## 4. Results and Discussions

### 4.1. Univariate Results

Table 1 presents the descriptive statistics of the main variables used in the analysis. The table contains two panels: panel A shows the statistics for the full sample, whereas panel B splits the study sample into family and non-family-controlled firms. The mean value of the annual bid–ask spread was 1.01, and the standard deviation was 2.07. Regarding panel B, the mean value of the bid–ask spread was slightly higher in non-family-controlled firms than family-controlled ones, at 1.01 and 0.97, respectively. In terms of ESG_SCORE, the average value for the full sample was 31.58, and those for ENV_SCORE, SOC_SCORE, and GOV_SCORE (refer to Appendix A for variable definitions) were 21.40, 34.68, and 54.98, respectively, showing that the sample firms performed better in the governance component. In panel B, the mean values of ESG_SCORE and its components (i.e., ENV_SCORE, SOC_SCORE, and GOV_SCORE) tended to be lower in family-controlled firms (28.27, 17.12, 32.23, and 53.45, respectively) than non-family-controlled ones (32.38, 22.38, 35.27, and 55.35, respectively). The highest scores were related to the GOV_SCORE component in both family- and non-family-controlled firms. With regards to the other variables, there were no large differences between family- and non-family-controlled firms in terms of size, leverage, ROA, and analysts, whereas the average growth in family-controlled firms was significantly lower than in non-family-controlled firms.

**Table 1.** Descriptive statistics.

| Panel A | Full Sample | | | | |
|---|---|---|---|---|---|
| **Variable** | **Mean** | **Median** | **St. Dev.** | **Min** | **Max** |
| SPREAD | 1.00 | 0.26 | 1.99 | 0.02 | 41.86 |
| ESG_SCORE | 31.58 | 30.17 | 11.74 | 3.31 | 69.42 |
| ENV_SCORE | 21.40 | 18.60 | 13.95 | 0.83 | 73.55 |
| SOC_SCORE | 34.68 | 33.33 | 12.92 | 3.51 | 84.21 |
| GOV_SCORE | 54.98 | 53.57 | 7.94 | 10.71 | 82.14 |
| SIZE | 6.71 | 6.53 | 1.84 | 1.17 | 12.93 |
| LEV | 0.18 | 0.14 | 0.21 | 0.00 | 2.99 |
| ROA | 0.04 | 0.05 | 0.14 | −0.67 | 0.49 |
| GROWTH | 0.36 | 0.03 | 2.04 | −0.99 | 16.98 |
| ANALYSTS | 9.57 | 7.00 | 8.76 | 0.00 | 50.00 |

| Panel B | Non-family-controlled firms | | | Family-controlled firms | | | Comparison of means |
|---|---|---|---|---|---|---|---|
| Variables | Mean | Median | St. Dev. | Mean | Median | St. Dev. | *t*-test |
| SPREAD | 1.01 | 0.23 | 2.07 | 0.97 | 0.32 | 1.67 | 0.447 |
| ESG_SCORE | 32.38 | 30.99 | 11.86 | 28.27 | 27.12 | 10.61 | 7.939 *** |
| ENV_SCORE | 22.38 | 19.83 | 14.14 | 17.12 | 13.95 | 12.23 | 7.958 *** |
| SOC_SCORE | 35.27 | 33.33 | 13.12 | 32.23 | 31.58 | 11.73 | 5.151 *** |
| GOV_SCORE | 55.35 | 53.57 | 8.11 | 53.45 | 53.57 | 7.04 | 5.398 *** |
| SIZE | 6.74 | 6.55 | 1.88 | 6.59 | 6.49 | 1.69 | 2.143 ** |
| LEV | 0.19 | 0.15 | 0.21 | 0.16 | 0.12 | 0.17 | 4.132 *** |
| ROA | 0.03 | 0.04 | 0.14 | 0.06 | 0.06 | 0.14 | −5.627 *** |
| GROWTH | 0.40 | 0.02 | 2.23 | 0.17 | 0.07 | 0.79 | 2.934 *** |
| ANALYSTS | 9.53 | 7.00 | 9.08 | 9.70 | 8.00 | 7.48 | −0.464 |

***, **, and * indicate statistical significance at the 1%, 5%, and 10% levels, respectively. The study sample consisted of 641 UK firms listed on the FTSE All-Share Index during the period 2010–2017 and included 2058 firm-year observations. See Appendix A for definitions of the variables.

### 4.2. Correlation Analysis

Table 2 shows the results of the correlation matrix of all the independent variables. The values show that the highest correlation was 0.67, between ANALYSTS and SIZE, followed by 0.61, between SIZE and ESG_SCORE. Based on Gujarati and Porter [82], there was no multicollinearity issue, since the correlation between the two variables was less than 80%.

**Table 2.** Correlation coefficients.

| Variable | (1) | (2) | (3) | (4) | (5) | (6) |
|---|---|---|---|---|---|---|
| (1) ESG_SCORE | 1.000 | | | | | |
| (2) SIZE | 0.615 * | 1.000 | | | | |
| (3) LEV | 0.109 * | 0.174 * | 1.000 | | | |
| (4) ROA | 0.035 | 0.135 * | −0.140 * | 1.000 | | |
| (5) GROWTH | 0.049 * | 0.130 * | 0.008 | −0.007 | 1.000 | |
| (6) ANALYSTS | 0.521 * | 0.677 * | 0.140 * | 0.199 * | 0.011 | 1.000 |

This table shows the Spearman correlation coefficient among the main variables. The study sample included 641 UK firms listed on the FTSE All-Share Index during the period 2010–2017. See Appendix A for definitions of the variables. * shows significance at the 0.01 level.

### 4.3. Multivariate Results

Table 3 shows the results obtained for the models. Model 1 explored the direct impact of sustainability reporting on the bid–ask spread, which is a proxy for IA. The ESG_SCORE variable showed a negative and significant coefficient (−0.00973; $p < 0.001$), which is consistent with Hypothesis 1, suggesting that sustainability reporting can play a complementary role in reducing the information gap that arises between firms and stakeholders. This finding can be explained by stakeholder theory, in which managers have a fiduciary duty towards all stakeholders instead of maintaining exclusive relationships with them [40]. Meeting the expectations of different stakeholder groups by actively committing to sustainability reporting can help to improve a company's reputation [41]. Therefore, it has been argued that reputation building is linked to higher-quality earnings reporting [42], which ultimately reduces IA [43]. This outcome confirms the evidence reported by [6,16].

**Table 3.** Sustainability reporting–IA relationship using the GMM model.

| | Model (1) | Model (2) |
|---|---|---|
| SPREAD | 0.770 *** | 0.712 *** |
| | (0.080) | (0.071) |
| ESG_SCORE | −0.010 *** | 0.002 |
| | (0.004) | (0.004) |
| FAMILY_ESG | | −0.009 ** |
| | | (0.005) |
| FAMILY | | 0.352 ** |
| | | (0.164) |
| SIZE | 0.0287 * | 0.070 |
| | (0.022) | (0.070) |
| LEV | 1.903 * | 1.212 ** |
| | (0.991) | (0.611) |
| ROA | −1.127 *** | −1.025 ** |
| | (0.403) | (0.435) |
| GROWTH | −0.019 | −0.290 * |
| | (0.013) | (0.175) |
| ANALYSTS | −0.008 * | −0.006 |
| | (0.004) | (0.005) |
| Constant | 0.033 ** | 0.476 *** |
| | (0.207) | (0.338) |
| Arellano–Bond test for AR(2) in first differences: | z = 0.23 Pr > z = 0.818 | z = 0.16 Pr > z = 0.876 |
| Hansen test of overid. restrictions: | chi2(5) = 7.46 Prob > chi2 = 0.189 | chi2(5) = 10.49 Prob > chi2 = 0.275 |

*, **, and *** indicate statistical significance error at the levels of 10%, 5%, and 1%, respectively. The table shows the results of the GMM regression. The study sample consisted of 641 UK firms listed on the FTSE All-Share Index during the period 2010–2017 and included 2058 firm-year observations. The dependent variable was bid–ask spread, whereas the independent variable was ESG_SCORE, which combines environmental, social, and governance scores (all variables are defined in Appendix A).

In Model 2, the moderating role of family-controlled firms was introduced to the relationship between sustainability reporting and IA, showing that it is moderated by

family-controlled firms ($-0.00947$; $p < 0.05$). From this, we can observe that the negative relationship was weakened by introducing family-controlled firms and became positive (FAMILY_ESG + FAMILY) ($0.352 - 0.00947 = 0.34253$). This finding supports the adverse selection effect theory [15], which argues that family owners are involved in day-to-day operations and usually hold higher managerial positions, so they have better access to information than minor shareholders. In this case, family owners, who are more informed investors, tend to exploit this information for private purposes. Accordingly, the moderating role of family-controlled firms weakens and decreases the IA caused by corporate social and environmental disclosures.

Moreover, this finding suggests that family-controlled firms, with their information advantage, may disguise their trading through small transactions to maximise their profits by buying at low ask prices and selling at high bid prices [61]. This manner of trading can be continued until any private information is fully revealed to the public, or for as long as the profit from this trading against less-informed investors is sufficient to cover any information acquisition costs [61]. Easley and O'hara [30] suggest that informed investors can adjust their portfolios because of the private information they possess, whereas less-informed investors cannot adjust their portfolios effectively due to a lack of such information, which increases IA by raising the risks faced by less-informed investors and consequently widens the bid–ask spread. Accordingly, family ownership tends to weaken any reduction in IA that is linked to sustainability reporting.

## 5. Additional Tests

### 5.1. Additional Test—Sustainability Reporting Components

Considering sustainability reporting dimensions independently (i.e., environmental, governance, and social scores) could result in a better understanding of management strategies and their mentality towards sustainability reporting activities. For instance, some studies argue that managers tend to target outsiders by focusing only on issues related to the environment and community. This targeted approach is known as the "cherry-picking strategy", as it does not apply a holistic CSR strategy. Accordingly, considering sustainability reporting as an aggregate score would not be sufficient to interpret managers' behaviours related to its application [83]. Therefore, we furthered our analysis by exploring the impact of each sustainability reporting component on IA in family- and non-family-controlled firms.

Models 1, 3, and 5 presented in Table 4 show the results for each ESG component for the full sample, indicating that ENV_SCORE, SOC_SCORE, and GOV_SCORE all contributed to reducing IA (coef. $-0.00581$, $p < 0.05$; coef. $-0.00399$, $p < 0.1$; coef. $-0.0119$, $p < 0.1$, respectively). Their contributions, however, were not similarly consistent for family-controlled firms. Models 2, 4, and 6 integrated family-controlled firms with each ESG component, showing that not all the pillars contributed to mitigating IA. Model 2 shows a positive and significant relationship between ENV_SCORE and SPREAD (coef. 0.0399; $p < 0.01$), while in Model 4, the social pillar negatively affected SPREAD (coef. $-0.00399$; $p < 0.1$). Finally, GOV_SCORE was positively related to SPREAD.

**Table 4.** Sustainability reporting components and IA.

| Variable | Model (1) | Model (2) | Model (3) | Model (4) | Model (5) | Model (6) |
|---|---|---|---|---|---|---|
| ENV_SCORE | −0.006 ** | 0.040 ** | | | | |
| | (0.003) | (0.020) | | | | |
| SOC_SCORE | | | −0.004 * | −0.041 *** | | |
| | | | (0.002) | (0.014) | | |
| GOV_SCORE | | | | | −0.012 * | 0.001 |
| | | | | | (0.006) | (0.004) |
| FAMILY | | 2.875 ** | | −5.059 *** | | 0.628 ** |
| | | (1.349) | | (1.798) | | (0.305) |
| FAMILY_ENV | | −0.145 ** | | | | |
| | | (0.069) | | | | |
| FAMIY_SOC | | | | 0.151*** | | |
| | | | | (0.054) | | |

**Table 4.** *Cont.*

| Variable | Model (1) | Model (2) | Model (3) | Model (4) | Model (5) | Model (6) |
|---|---|---|---|---|---|---|
| FAMIY_GOV | | | | | | −0.0105 * |
| | | | | | | (0.005) |
| SIZE | −0.004 ** | −0.093 | −0.021 | 0.057 | 0.001 | −0.019 |
| | (0.026) | (0.107) | (0.024) | (0.068) | (0.043) | (0.093) |
| LEV | 0.170 | 0.388 | 0.623 | 2.015 *** | 0.799 | 1.731 *** |
| | (0.383) | (0.372) | (0.523) | (0.511) | (0.492) | (0.646) |
| ROA | −0.731 *** | −1.214 *** | −0.793 ** | −0.781 *** | −0.839 ** | −0.979 ** |
| | (0.275) | (0.372) | (0.349) | (0.294) | (0.397) | (0.463) |
| GROWTH | −0.007 | 0.012 | −0.011 | 0.049 ** | −0.019 | −0.124 |
| | (0.013) | (0.019) | (0.014) | (0.024) | (0.014) | (0.204) |
| ANALYSTS | −0.005 | −0.026 | −0.004 | −0.018 | −0.004 | −0.007 |
| | (0.006) | (0.020) | (0.007) | (0.013) | (0.010) | (0.015) |
| Constant | 0.186 ** | 0.251 * | 0.269 * | 0.474 ** | 0.613 * | −0.094 * |
| | (0.153) | (0.687) | (0.147) | (0.5) | (0.358) | (0.642) |
| Arellano–Bond test for AR(2) in first differences: | z = 0.62 Pr > z = 0.535 | z = 0.37 Pr > z = 0.714 | z = 0.61 Pr > z = 0.542 | z = −0.05 Pr > z = 0.962 | z = 0.45 Pr > z = 0.654 | z = 0.33 Pr > z = 0.743 |
| Hansen test of overid. restrictions: | chi2(8) = 11.68 Prob > chi2 = 0.166 | chi2(5) = 2.90 Prob > chi2 = 0.716 | chi2(8) = 12.13 Prob > chi2 = 0.146 | chi2(5) = 7.46 Prob > chi2 = 0.189 | chi2(8) = 11.51 Prob > chi2 = 0.175 | chi2(8) = 12.05 Prob > chi2 = 0.149 |

*, **, and *** indicate statistical significance at the levels of 10%, 5%, and 1%, respectively. The table shows the results of the GMM regressions for each ESG component separately in family- and non-family-controlled firms. The study sample consisted of 641 UK firms listed on the FTSE All-Share Index during the period 2010–2017 and included 2058 firm-year observations. The dependent variable was bid–ask spread, whereas the independent variables were ENV_SCORE, SOC_SCORE, and GOV_SCORE. All variables are defined in Appendix A.

### 5.2. Additional Tests—Alternative Measurement of Sustainability Reporting

CSRhub scoring was used as an alternative to the Bloomberg ESG scoring. CSRhub is a tool that provides access to sustainability and corporate social responsibility (CSR) ratings, with information on more than 18,424 companies from 136 different industries in 141 countries. The scoring system relies on twelve subcategories (indicators), which are grouped into four main categories: employees, community, environment, and governance (see Appendix B). To date, CSRhub has mapped more than 5000 data elements related to sustainability reporting, sorting each element into one or more subcategories and, if it does not fall under any of the twelve indicators, categorising it as a special issue. The data are aggregated and normalised from eight leading sustainability reporting analysts (ASSET4/Thomson Reuters; Trucost; ET Index; Carbon Disclosure Project (CDP); Vigeo EIRIS; IW Financial; MSCI (Risk Metrics IVA and Impact Monitor); RepRisk; Governance Metrics International/Corporate Library; and MSCI (Carbon Tracker, ESG Intangible Value Assessment, and ESG Impact Monitor)). Therefore, CSRhub represents the world's largest and most comprehensive dataset on sustainability and CSR.

After aggregating data from the abovementioned entities, each data source was converted into a rating from 0 to 100. The scores from different datasets were then compared for the same company, allowing variations (biases) between sources to be determined and adjusted/eliminated to create more consistent ratings, making the data relatively objective and not based solely on self-reported measures. Finally, ratings were obtained for each subcategory and aggregated to a category level. CSRHub has been used and recommended by Bu, Liu [84] and Cruz, Larraza–Kintana [70].

The results in Table 5 Model 1 show that the relationship between sustainability reporting and IA remained negative and significant when considering the full sample. In Model 2, however, the values show that the ESG_IA relationship became positive in family-controlled firms, which is consistent with the earlier findings.

**Table 5.** CSR performance–IA relationship.

|  | Model (1) | Model (2) |
|---|---|---|
| SPREAD | 1.035 *** | 0.674 *** |
|  | (0.293) | (0.128) |
| ESG_CSRHUB | −0.021 ** | 0.015 |
|  | (0.009) | (0.021) |
| FAMILY |  | 5.761 ** |
|  |  | (2.403) |
| FAMILY_ESGHUB |  | 0.097 ** |
|  |  | (0.040) |
| SIZE | 0.726 * | −0.552 *** |
|  | (0.396) | (0.200) |
| LEV | 1.916 | 20.120 *** |
|  | (1.591) | (4.114) |
| ROA | 0.359 | 4.649 * |
|  | (2.822) | (2.594) |
| GROWTH | 0.046 | −1.262 |
|  | (0.053) | (1.257) |
| MKTOBV | −0.376 ** | 1.507 ** |
|  | (0.184) | (0.732) |
| ANALYSTS | −0.072 * | 0.005 |
|  | (0.038) | (0.023) |
| Constant | −3.244 | −1.424 |
|  | (2.647) | (1.665) |
| Arellano–Bond test for AR(2) in first differences: | z = −0.17 Pr > z = 0.865 | z = −1.36 Pr > z = 0.174 |
| Hansen test of overid. restrictions: | chi2(5) = 2.42 Prob > chi2 = 0.789 | chi2(9) = 14.32 Prob > chi2 = 0.111 |

*, **, and *** indicate statistical significance at the levels of 10%, 5%, and 1%, respectively. The table shows the results of the GMM regressions for family- and non-family-controlled firms. The study sample consisted of 641 UK firms listed on the FTSE All-Share Index during the period 2010–2017 and included 2058 firm-year observations. The dependent variable was the bid–ask spread, whereas the independent variable was ESG performance, adopted from the CSRhub database (all variables are defined in Appendix A).

## 6. Conclusions

Despite the growing importance of sustainability reporting for investors and other stakeholders, the majority of studies have focused more on financial disclosures and their influence on IA, whereas there is little empirical evidence [3,6,15,16] to show whether, and in what way, sustainability reporting (an example of non-financial disclosure) can complement financial disclosures in reducing IA problems [3]. Therefore, this study examines the influence of sustainability reporting on IA, taking into account the influence of each sustainability reporting component (environmental, social, and governance) independently. It also addresses how family-controlled firms, as informed investors, can moderate the sustainability reporting–IA nexus, since the key owners exert significant influence over a firm's investment decisions by suggesting and voting on strategic plans for the firm [85].

The findings show that sustainability reporting, as an aggregate score, reduces IA. This finding supports hypothesis H1, which proposed that managers have a fiduciary duty towards all stakeholders and, therefore, meeting the expectations of different stakeholder groups by actively committing to sustainability reporting can help improve a company's reputation [41]. Consequently, reputation building is argued to be linked to higher-quality earnings reporting [42], which ultimately reduces IA [43]. Kim, Park [44] and Scholtens and Kang [45] add that CSR creates a general atmosphere that inspires managers to develop a public-responsibility-oriented mentality, which subsequently encourages the issuance of more transparent financial reporting and the meeting of stakeholder expectations. The second main finding was that the negative relationship between sustainability reporting and IA weakens and even becomes positive in family-controlled firms. This supports the adverse selection perspective of Hypothesis H2, in which family-controlled firms take advantage of the information they have access to at the cost of less-informed investors.

Finally, in the models investigating the direct impact of each ESG pillar (environmental, social, and governance scores) on IA, all components still showed a negative influence on the bid–ask spread. More specifically, disclosures about environmental issues tended to have a slightly stronger influence on IA compared to social and governance scores. However, the moderating role of family-controlled firms in the three models weakened the negative influence of environmental and governance scores and strengthened the negative impact of social scores on IA. This outcome indicates that family-controlled firms follow a cherry-picking CSR strategy rather than applying a holistic approach and focus more on information related to social activities.

Generally, these findings provide a meaningful insight into the CSR strategies followed by family-controlled firms; thus, they could help British regulators improve corporate governance rules related to ownership structure, since family-controlled firms can exploit their private-information access privileges. For investors, our findings provide evidence of the importance of CSR information in improving a firm's reputation and increasing its value. The findings also reveal to investors that the role of family-controlled firms is a double-edged sword. On the one hand, family-controlled firms can minimise the first type of agency problem, the principal–agent problem. On the other hand, family-controlled firms can exacerbate the second type of agency conflict, which exists between the majority and minority shareholders.

In terms of theoretical and academic implications, as mentioned in the previous paragraph, our investigation of the interaction between family-controlled firms and each CSR element provided evidence of the CSR disclosure strategies employed by such companies. Accordingly, studies of family-controlled firms should consider their heterogeneity and should not implicitly assume that family-controlled firms are ethically driven.

Overall, this study contributes to the existing literature in several ways. First, while previous studies have tested the moderating role of firm characteristics [6], equity risk [16], and institutional ownership [15] on the sustainability reporting–IA relationship, other ownership types have been overlooked. Therefore, this study contributes to the body of knowledge by addressing the moderating role of family-controlled firms in the relationship between sustainability reporting and IA. The reason for choosing family firms was that they are argued to have unique characteristics that distinguish them from other companies [24,25]. Moreover, family ownership is considered to be the most prevalent ownership type around the world [26–28].

Second, the study extends the methodological approaches of previous studies by using the generalised method of moments (GMM) model. The causal association between sustainability reporting and IA could be endogenous because of managerial policies and other factors that result in simultaneity and reverse causality. Therefore, if sustainability reporting and IA are simultaneously determined, ordinary least squares (OLS) will not be accurate. Consequently, based on Arellano and Bond [29], the GMM model was useful in addressing these issues and controlling for heterogeneity. Third, in addition to using aggregate ESG scores, this study also individually tested the influence of each ESG element (environmental, social, and governance). We thus provide a clearer explanation of management strategies and behaviour related to undertaking different sustainability reporting activities.

The study has some limitations that could be addressed in future research. First, the variable that represented family-controlled firms was a dummy variable, taking a value of 1 if family ownership exceeded 5% and/or there were two or more board members from the family, and a value of 0 otherwise. Relying only on a categorical variable (with one cut-off point/threshold) e.g., [53,86,87] may be insufficient to draw a clear conclusion about family ownership behaviour. Therefore, future studies could use continuous variables with different cut-off points, which would better reflect the level of family influence and involvement e.g., [81,88,89]. Second, the study used bid–ask spread as a proxy for IA, which is one of the common measures used in previous studies. It would be interesting if future studies verified the findings of this research by using different proxies (e.g., stock

liquidity (trading volume), price volatility, market-to-book ratio, the accuracy of analysts' forecasts, and the price impact measure). Third, in spite of the fact that the study duration of 2010–2017 was split between the period when sustainability reporting was voluntary (that is, prior to 2013) and the period when it became mandatory (after 2013) it tested ESG disclosures without differentiating between voluntary and mandatory ones. Studying the effect of voluntary and mandatory disclosures separately could provide further explanation of management behaviour. Therefore, future studies could investigate the relationship between sustainability reporting and EM before and after the mandating of ESG disclosures in the UK.

**Author Contributions:** Conceptualization, A.R.A.N. and R.M.; Formal analysis, A.R.A.N. and H.Z.; Investigation, S.K. and H.Z.; Methodology, R.M.; Project administration, A.R.A.N. and R.M.; Software, R.M.; Supervision, H.Z.; Validation, S.K. and R.M.; Visualization, S.K.; Writing—original draft, R.M.; Writing—review & editing, A.R.A.N. All authors have read and agreed to the published version of the manuscript.

**Funding:** This research received no external funding.

**Institutional Review Board Statement:** Not applicable.

**Informed Consent Statement:** Not applicable.

**Conflicts of Interest:** The authors declare no conflict of interest.

## Appendix A. Variable Definitions

**Table A1.** Variable definitions.

| | |
|---|---|
| SPREAD | The annual average percentage of the daily bid-ask spread to the closing price. |
| ESG | Total sustainability reporting score composite from the Bloomberg database, which combines four dimensions (environment, community, employees, and governance). |
| ENV_SCORE | Total environmental score (source: Bloomberg). |
| SOC_SCORE | Total social score, which includes both community and employee scores (source: Bloomberg). |
| GOV_SCORE | Total governance score (source: Bloomberg). |
| FAMILY | Dummy variable equal to 1 for family-controlled firms and 0 otherwise (source: FAME). |
| FAMILY_ESG | The interaction between family-controlled firms and AEM. |
| SIZE | Natural logarithm of total assets. |
| LEV | The leverage ratio, measured as long-term debt scaled by total assets. |
| ROA | The return on assets ratio, measured as income before extraordinary items scaled by lagged total assets. |
| GROWTH | Sales growth rate. |
| ANALYSTS | The number of analysts following the company. |

## Appendix B. Sustainability Reporting Categories and Subcategories

**Table A2.** Sustainability reporting categories and subcategories.

| Community | Employees | Environment | Governance |
|---|---|---|---|
| Community Development and Philanthropy | Compensation and Benefits | Energy and Climate Change | Board |
| Product | Diversity and Labour Rights | Environmental Policy and Reporting | Leadership Ethics |
| Human Rights and Supply Chain | Training, Health, and Safety | Resource Management | Transparency and Reporting |

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
