# Peer review of "The Role of Sustainability Reporting in Reducing Information Asymmetry: The Case of Family- and Non-Family-Controlled Firms"

_sustainability, doi:10.3390/su14116644_

Round 1

Reviewer 1 Report

Dear authors,

The topic of this research is quite interesting. While reviewing the manuscript, I found a small mistake that the references are not properly formatted according to the MDPI's reference standard. Therefore, I recommend the authors examine the whole manuscript and reformat references based on Sustainability's guidelines.

  1. Introduction
  • The authors described the background of this study by showing the importance of sustainability reporting in lines 30 to 37. Also, the authors pointed out the research gap in lines 38 to 43. However, these explanations fail to explain why the research on the relationship between sustainability reporting and IA is necessary. Therefore, I recommend the authors further provide a convincing explanation of the necessity of such research.
  • On page 2, lines 58 to 111 describe the analysis result. I suggest the authors relocate these contents to the conclusion. I also noticed that the authors had omitted discussion on research questions in Section 1. Discussing research questions in Section 1 is important because this helps readers understand the manuscript's contents and the authors' intention.

2.2. The role of family-controlled firms

  • Considering its contents, I think the current title needs some modification.
  • (As Is) The role of family-controlled firms 
  • (To Be) The Influence of firm ownership on information asymmetry

6. Conclusion

  • In Section 6, lines 479 to 501 summarize the overall analysis results. There are some repetitions in these two paragraphs. Therefore I recommend the authors review and revise the conclusion.
  • In lines 515 to 519, the authors depicted that investigating the relationship between sustainability reporting and IA of UK companies is one of the contributions of the manuscript. I believe that examining UK cases can differentiate this manuscript from other studies; even so, it cannot be one of the contributions of the manuscript. 

Author Response

First and foremost, thank you for your valuable and constructive comments. Each comment has served to augment and improve the quality of the manuscript. We have addressed the comments in detail. As a result, substantial changes have been made directly to the manuscript, as well as recorded all the changes made using tracked changes tool. Considering the aforesaid, we hope that the paper in its revised form will match your expectations.

Reviewer Comments

I found a small mistake that the references are not properly formatted according to the MDPI's reference standard. Therefore, I recommend the authors examine the whole manuscript and reformat references based on Sustainability's guidelines.

Author responses: the references are now formatted based on the MDPI’s reference standards.

  1. Introduction
  • The authors described the background of this study by showing the importance of sustainability reporting in lines 30 to 37. Also, the authors pointed out the research gap in lines 38 to 43. However, these explanations fail to explain why the research on the relationship between sustainability reporting and IA is necessary. Therefore, I recommend the authors further provide a convincing explanation of the necessity of such research.

Author response: we agree with you, the more convincing discussion provided in the literature review instead of the introduction, indeed. Therefore, we revised the whole introduction and tried to further highlight the research importance and gap (please refer to lines 37-51). We are happy to further modify it, if needed.

  • On page 2, lines 58 to 111 describe the analysis result. I suggest the authors relocate these contents to the conclusion. I also noticed that the authors had omitted discussion on research questions in Section 1. Discussing research questions in Section 1 is important because this helps readers understand the manuscript's contents and the authors' intention.

Author response: thank you for mentioning these points. In terms of the analysis results, please note that our results were introduced in lines 90-111, whereas the previous lines (58-89) talk about the research contributions. However, we further read the mentioned paragraphs and tried to refine them in a more readable way.

In terms of the research question, we tried to introduce it indirectly in lines 48 and 48 as follows, “…whereas a little is known on whether, and in what way, Sustainability Reporting, as a part of non-financial disclosures…”.

2.2. The role of family-controlled firms

  • Considering its contents, I think the current title needs some modification.
  • (As Is) The role of family-controlled firms 
  • (To Be) The Influence of firm ownership on information asymmetry

Author response: we replaced the title to become “The Influence of firm ownership on information asymmetry” as recommended.

  1. Conclusion
  • In Section 6, lines 479 to 501 summarize the overall analysis results. There are some repetitions in these two paragraphs. Therefore, I recommend the authors review and revise the conclusion.

Author response: thank you for your comment. We just need to clarify that the first paragraph you mentioned talks about our main finding when we used an aggregate measure of the ESG (single score), whereas the other paragraph talks about the findings of each environmental, social and governance categories, separately.

  • In lines 515 to 519, the authors depicted that investigating the relationship between sustainability reporting and IA of UK companies is one of the contributions of the manuscript. I believe that examining UK cases can differentiate this manuscript from other studies; even so, it cannot be one of the contributions of the manuscript. 

Author response: thank you for your suggestion, we removed it from both the conclusion and the introduction.

Best regards,

Authors

Reviewer 2 Report

Dear Authors,

The manuscript is good. You tried to investigate  the link between Sustainability Reporting and information 13 asymmetry in family and non-family-controlled firms for a sample of 641 UK listed firms in the 14 FTSE all-share index during the period 2010-2017.

Please fix the references and citations. For example:

In recent years, Sustainability Reporting has become an area for market participants 28
(Cho et al., 2012).

Please use the journal style. 

Add: Theoretical contribution, practical contribution and managerial implications. 

Good luck!!!!

Author Response

First and foremost, thank you for your valuable and constructive comments. Each comment has served to augment and improve the quality of the manuscript. We have addressed the comments in detail. As a result, substantial changes have been made directly to the manuscript, as well as recorded all the changes made using tracked changes tool. Considering the aforesaid, we hope that the paper in its revised form will match your expectations.

Reviewer Comments:

  1. The manuscript is good. You tried to investigate the link between Sustainability Reporting and information asymmetry in family and non-family-controlled firms for a sample of 641 UK listed firms in the FTSE all-share index during the period 2010-2017.

Please fix the references and citations. For example: In recent years, Sustainability Reporting has become an area for market participants (Cho et al., 2012).

Please use the journal style.

Author response: thank you for your constructive comments, we revised the whole manuscript and fixed this referencing issue.

  1. Add: Theoretical contribution, practical contribution and managerial implications. 

Author response: in addition to the implications and contributions we mentioned in both the introduction and conclusion sections, we further highlighted the theoretical, academic and managerial implications in the conclusion section as suggested (please refer to lines 500-516).

Best regards,

Authors

Reviewer 3 Report

The paper deals with the increasingly common problem of sustainable development reporting. This type of reporting represents non-financial information and can play a complementary role to financial disclosures in mitigating information asymmetry issues.

In the literature review section, I would suggest adding information on the principles of financial reporting. Financial reporting is regulated in legal acts, international accounting and financial reporting standards, individual regulations in force on individual stock exchanges, the statement: “financial disclosures are more likely to be mandatory”, it is too laconic. I would suggest adding the concept of standardizing non-financial reporting. In the case of financial reporting, there are such regulations, which often constitute the basis for assessing the quality of the prepared report, for example during auditing activities for the benefit of the company and its stakeholders.

Author Response

First and foremost, thank you for your valuable and constructive comments. Each comment has served to augment and improve the quality of the manuscript. We have addressed the comments in detail. As a result, substantial changes have been made directly to the manuscript, as well as recorded all the changes made using tracked changes tool. Considering the aforesaid, we hope that the paper in its revised form will match your expectations.

Reviewer Comments:

The paper deals with the increasingly common problem of sustainable development reporting. This type of reporting represents non-financial information and can play a complementary role to financial disclosures in mitigating information asymmetry issues.

In the literature review section, I would suggest adding information on the principles of financial reporting. Financial reporting is regulated in legal acts, international accounting and financial reporting standards, and individual regulations in force on individual stock exchanges, the statement: “financial disclosures are more likely to be mandatory”, it is too laconic. I would suggest adding the concept of standardizing non-financial reporting. In the case of financial reporting, there are such regulations, which often constitute the basis for assessing the quality of the prepared report, for example during auditing activities for the benefit of the company and its stakeholders.

Author response: thank you for this important suggestion. The concept of standardizing non-financial reporting has been mentioned indirectly as a limitation in our study and suggestions for future studies (please refer to lines 807-813). We recommended future studies to divide the study sample into two periods, a period before 2013, which is the period before mandating CSR “compulsory CSR” reporting in the UK, and a period after 2013. Doing so will give an indicator of how mandatory CSR reporting can be perceived by the market participants compared with voluntary CSR reporting. Furthermore, it could explain how family-controlled firms as strategic shareholders behave under institutional pressures. This is the reason for not discussing the standardisation of CSR reporting in the literature review.

In terms of the financial disclosures, as you noticed in the literature, we just gave a generic idea about financial disclosures, since our focus is on CSR reporting only.

We reviewed the following papers to get a better idea about standardised non-financial disclosures, and we confirm that our next projects will give this issue more weight, especially with the introduction of IFRS Sustainability disclosure standards.

  • https://www.legislation.gov.uk/ukdsi/2013/9780111540169

  • https://www.ifrs.org/news-and-events/news/2022/03/issb-delivers-proposals-that-create-comprehensive-global-baseline-of-sustainability-disclosures/#:~:text=IFRS%20Sustainability%20Disclosure%20Standards%20are,meet%20broader%20stakeholder%20information%20needs.

  • Adams, C.A. and Mueller, F., 2022. Academics and policymakers at odds: the case of the IFRS Foundation Trustees’ consultation paper on sustainability reportingSustainability Accounting, Management and Policy Journal, (ahead-of-print).

  • Krawczyk, P., 2021. Non-Financial Reporting—Standardization Options for SME SectorJournal of Risk and Financial Management14(9), p.417.

  • Tamvada, M., 2020. Corporate social responsibility and accountability: a new theoretical foundation for regulating CSRInternational Journal of Corporate Social Responsibility5(1), pp.1-14.

  • Ioannou, I. and Serafeim, G., 2017. The consequences of mandatory corporate sustainability reportingHarvard Business School research working paper, (11-100).

  • Krištofík, P., Lament, M. and Musa, H., 2016. The reporting of non-financial information and the rationale for its standardisation.

Best regards,

Authors

Round 2

Reviewer 1 Report

This manuscript has been improved a lot, and I think now it is acceptable for publication in Sustainability.

Reviewer 2 Report

The revised version is okay. The authors are required to copy-edit the entire manuscript to ensure the quality of the manuscript.